crystallography/organic chemistry/physical chemistry

electrosynthesis, furan oxidation, 2,5-diacetoxy-2,5-dihydrofuran, 2,5-dicarboxy-2,5-dihydrofurans, stereoselectivity

**Author for correspondence:**
Mark D. Symes
e-mail: mark.symes@glasgow.ac.uk

This article has been edited by the Royal Society of Chemistry, including the commissioning, peer review process and editorial aspects up to the point of acceptance.

# Towards a better understanding of the electrosynthesis of 2,5-dicarboxy-2,5-dihydrofurans: structure, mechanism and influence over stereochemistry

Michael A. Shipman, Stephen Sproules, Claire Wilson and Mark D. Symes

WestCHEM, School of Chemistry, University of Glasgow, University Avenue, Glasgow G12 8QQ, UK

(iD) MDS, 0000-0001-8067-5240

2,5-Dicarboxy-2,5-dihydrofurans are key constituents of a number of natural products and have roles as intermediates in the formation of other such compounds of interest. Typically, these species are synthesized using toxic Pb(IV) salts. Electrochemical syntheses of 2,5-diacetoxy-2,5-dihydrofuran that do not require the use of lead have been reported, but a general lack of experimental detail has prevented these procedures from being more widely adopted. Moreover, no electrochemical study has yet reported the ratio of *cis* and *trans* isomers produced. Herein, we compare the chemical, lead-based route to 2,5-diacetoxy-2,5-dihydrofuran with a fully described electrosynthesis method. In doing so, we have discovered that the *cis* and *trans* isomers of this compound were previously incorrectly assigned in the literature, an error that we correct by obtaining the crystal structure of *cis*-2,5-diacetoxy-2,5-dihydrofuran. This allows the ratios of the isomers as prepared by the chemical (2 : 1 *cis* : *trans*) and electrochemical (7 : 5 *cis* : *trans*) methods to be obtained. Through experimental and computational insights, we propose a mechanism for the electrochemical synthesis of 2,5-dicarboxy-2,5-dihydrofurans and go some way towards validating this mechanism by synthesizing 2,5-dibutoxy-2,5-dihydrofuran electrochemically for the first time. We hope that these findings will provide some greater clarity to the literature surrounding the electrosynthesis and potential applications of 2,5-dicarboxy-2,5-dihydrofurans.

# 1. Introduction

2,5-Dicarboxy-2,5-dihydrofurans can be found in natural products such as aplysulfurin [1], the thuridillins [2] and the prunolides [3]. They have also been used as building blocks in the preparation of species such as aflatoxin B$_1$ [4], and related moieties can be found in the dendrillolides [5] and aplyviolene [6]. Moreover, 2,5-diacetoxy-2,5-dihydrofuran has been employed as a precursor to various butenolides [7], nucleosides [8], and molecules with potential pharmaceutical [9–11] and technological applications [12].

In the majority of cases, these 2,5-dicarboxy-2,5-dihydrofurans are prepared by reaction of furan with the appropriate tetracarboxy salt of lead according to the general method of Elming & Clauson-Kaas [13]. An attractive route for obtaining such products without the use of toxic lead salts is the electrochemical oxidation of furan in the presence of acetate to give 2,5-diacetoxy-2,5-dihydrofuran. This approach was first reported by Wilson & Lippincott [14], and subsequently developed firstly by Baggaley & Brettle [15] and then by Shono *et al.* [7]. However, little attention seems to have been paid to this electrochemical route in recent years, aside from a report by Horii *et al.* [16] describing the electrosynthesis of 2,5-diacetoxy-2,5-dihydrofuran in a thin-layer flow cell. This last work is also the only report of the electrochemical synthesis of a 2,5-dicarboxy-2,5-dihydrofuran with a comprehensive and detailed experimental procedure. To date, all these electrosynthesis methods have been conducted under constant current conditions.

A complicating factor with both the chemical (Pb-promoted) and electrochemical methods for the preparation of 2,5-dicarboxy-2,5-dihydrofurans is the fact that two isomers of the product are possible, one where the carboxy substituents are *trans* to one another and one where they have a *cis* arrangement (figure 1). Hitherto, it has not proved possible to control which isomer forms to any significant degree using either the chemical or electrochemical methods. Moreover, even in the case of the simplest of these species, 2,5-diacetoxy-2,5-dihydrofuran, an unambiguous identification of the two isomers remains elusive. Against this background, definitive assignment of the two stereoisomers of 2,5-diacetoxy-2,5-dihydrofuran is essential if 2,5-dicarboxy-2,5-dihydrofurans are to realize their potential as building blocks in organic synthesis. Meanwhile, any insights into affording a greater control over the stereoselectivity of this reaction would also be most useful.

Herein, we describe our attempts to determine the stereoselectivity of the electrochemical synthesis of 2,5-diacetoxy-2,5-dihydrofuran under batch conditions and to provide a more detailed methodology for this process than has been reported to date. In the course of this study, we isolated and purified the mixture of 2,5-diacetoxy-2,5-dihydrofuran isomers, which in turn allowed us to assign for the first time and unambiguously which isomer is which. Contrary to our expectations and the previous literature reports [4,17,18] we show that the crystalline isomer is in fact the *cis* isomer (and not the *trans* isomer as had been assumed). This fact obviously has a major bearing on trying to gain control over which isomer forms and also has implications for using the products of this reaction in subsequent chemical syntheses. As a direct result of this discovery, we propose a route by which carboxylate groups add to electrochemically oxidized furans and validate this model by reversing the stereoselectivity for the *cis* isomer generally displayed when using acetate, by using the bulkier substituent butyrate to give predominantly the *trans* product. We hope that these findings will allow deeper insight into the electrosynthesis (and potential applications) of 2,5-dicarboxy-2,5-dihydrofurans.

# 2. Experimental section

## 2.1. General experimental remarks

Unless otherwise stated, all syntheses were conducted under nitrogen in air- and moisture-free solvents obtained from a commercial solvent purification system. Water used was of 'ultra-pure' grade (18.2 MΩ-cm resistivity) and dispensed from an SG Ultraclear TWF UV device. Sodium acetate (≥99%), sodium butyrate (98%), butyric acid (≥99%), furan (≥99%), diethyl ether (≥99.8%) and lead(IV) acetate (96% + 5–10% glacial acetic acid) were supplied by Sigma Aldrich. Acetonitrile (≥99%), sodium carbonate (≥99.9%) and nitric acid (70%) were supplied by Fisher Scientific. Acetic acid (99.9%) was purchased from VWR. Liquid nitrogen was supplied by BOC. HCl was obtained from Honeywell Fluka.

All $^1$H and $^{13}$C nuclear magnetic resonance (NMR) spectra were recorded on a Bruker AV 400 instrument (unless otherwise stated), at a constant temperature of 300 K. Chemical shifts are reported in parts per million from low to high field. Standard abbreviations indicating multiplicity were used as follows: m, multiplet; s, singlet. Melting points were gauged using a Stuart Scientific SMP10 melting point apparatus. Experiments performed at 'room temperature' were carried out at 20°C. Electrochemical experiments were performed as

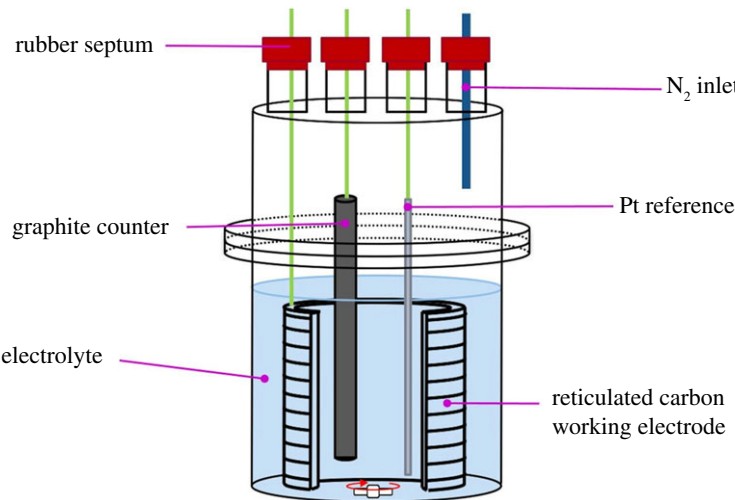

**Figure 1.** The *cis* and *trans* isomers of 2,5-diacetoxy-2,5-dihydrofuran discussed in this paper. Italic letters on these structures correspond to the $^1$H NMR signal assignments in the Experimental section.

**Figure 2.** A typical set-up used for the bulk electrolysis experiments reported in this paper.

below. Chemical synthesis of 2,5-diacetoxy-2,5-dihydrofuran was carried out by adapting the method of Holzapfel & Williams [4] (in particular the work-up step), which allowed crystals of suitable quality for single crystal X-ray diffraction to be obtained (see below). The data which underpin this work are available via Enlighten, the University of Glasgow's open access online data repository [19].

## 2.2. Electrochemical methods

Electrochemical studies were performed in a single-chamber cell in a three-electrode configuration using a CH Instruments CHI600 series potentiostat. Bulk electrolyses were performed using a large surface area reticulated vitreous carbon electrode (Alvatek Ltd, UK) as the working electrode, and a Pt wire pseudo-reference electrode and a graphite rod (99.9995%; Alfa Aesar) as the counter electrode. Working electrodes were washed with water, followed by aqua regia (3 : 1 mixture of conc. $HCl : HNO_3$), then rinsed with water and acetone prior to use. Counter electrodes were sanded using 800 grit sandpaper to remove surface contamination. They were then sonicated for 1 min in water to remove any fine graphite particles, followed by rinsing with water, then acetone. The electrochemical cells were rinsed with aqua regia, water and acetone prior to use. Electrodes and glassware were dried under a stream of $N_2$ to facilitate removal of any remaining acetone.

A typical cell set-up is shown in figure 2. The use of a sealable cell allowed the entire experiment to be conducted under an inert atmosphere of nitrogen. The electrolyte was first transferred to the cell under a constant stream of $N_2$, and the cell was then lowered into an ice bath and the temperature allowed to equilibrate before furan was injected through a septum into the electrolyte. Bulk electrolyses were performed under potentiostatic control with stirring at a potential of 3 V versus Pt (approx. 2 V versus a standard hydrogen electrode, gauged by obtaining the position of the standard potassium ferricyanide/ferrocyanide couple [20] in the electrolyte).

## 2.3. Electrosynthesis and purification of 2,5-diacetoxy-2,5-dihydrofuran

To a suspension of sodium acetate (4.00 g, 48.8 mmol) in acetonitrile (20 ml) was added 80 ml acetic acid. The mixture was stirred until all the solids were dissolved, then transferred to the electrochemical cell described in figure 2, cooled to 0°C in an ice bath and the solution degassed for 1 h using a constant stream of nitrogen. After this time, furan (0.50 ml, 0.468 g, 6.88 mmol) was injected into the electrolyte under

nitrogen. The constant flow of nitrogen was then immediately switched to a static atmosphere to limit furan evaporation. Bulk electrolysis was then initiated at a potential of 3 V (versus Pt), and the charge passed was monitored. After bulk electrolysis, the reaction mixture was added to water (approx. 300 ml) and extracted from aqueous solution with an equal volume of dichloromethane. The organic phase was then washed with water to remove any residual acetic acid. Concentration of the organic phase *in vacuo* resulted in a viscous brown oil. This oil was added to an equal volume of diethyl ether, precipitating a dark brown solid (which was filtered off and discarded) and a yellowish solution. Concentration of this solution *in vacuo* yielded a yellow oil, the ${}^1$H NMR spectrum of which showed it to consist almost exclusively of a mixture of the *cis* and *trans* isomers of 2,5-diacetoxy-2,5-dihydrofuran, and the analytical data of which were in agreement with those reported previously (save for the assignment of the *cis* and *trans* isomers; see below) [4]. We therefore give the assigned ${}^1$H NMR data as follows: *cis*-2,5-diacetoxy-2,5-dihydrofuran (CDCl$_3$, 400 MHz), $\delta = 6.64$ (s, 2H, *cis*-H$_a$), 6.15 (s, 2H, *cis*-H$_b$), 1.99 (s, 6H, *cis*-H$_c$); *trans*-2,5-diacetoxy-2,5-dihydrofuran (CDCl$_3$, 400 MHz), $\delta = 6.86$ (s, 2H, *trans*-H$_a$), 6.13 (s, 2H, *trans*-H$_b$), 1.97 (s, 6H, *trans*-H$_c$). Assignments of signals to groups of protons are based on two-dimensional (correlation spectrum (COSY) and heteronuclear multiple quantum coherence (HMQC)) spectra and letter codes correspond to those shown in figure 1. Integrations are only to be compared with other signals in the same isomer (see below for a discussion of the relative amounts of *cis*- and *trans*-2,5-diacetoxy-2,5-dihydrofuran formed under various conditions). ${}^{13}$C NMR for *cis*-2,5-diacetoxy-2,5-dihydrofuran (CDCl$_3$, 100 MHz), $\delta = 169.6$ (**C**=O), 130.8 (**C**=C in furan ring), 99.8 (sp$^3$ **C** in furan ring), 20.9 (−**C**H$_3$). ${}^{13}$C NMR for *trans*-2,5-diacetoxy-2,5-dihydrofuran (CDCl$_3$, 100 MHz), $\delta = 169.7$ (**C**=O), 131.1 (**C**=C in furan ring), 101.1 (sp$^3$ **C** in furan ring), 20.8 (−**C**H$_3$). Sample one-dimensional NMR spectra are shown in the electronic supplementary material, along with COSY and HMQC spectra (electronic supplementary material, figures S1–S7).

## 2.4. Separation of isomers of 2,5-diacetoxy-2,5-dihydrofuran

To separate the isomers of 2,5-diacetoxy-2,5-dihydrofuran, the yellow oil of the mixture of isomers was dissolved in an equal volume of diethyl ether. The resulting solution was then cooled by immersion in a bath of liquid nitrogen, causing pale yellow crystals to form. At this point, the supernatant solution was decanted and replaced with fresh (room temperature) diethyl ether, which had the effect of re-dissolving the crystals. This solution was then again cooled by immersion in liquid nitrogen, causing crystallization. The supernatant was again removed. By repeated cool–crystallize–decant cycles, all the colour could be removed from the crystals. The products of this process were thus a yellow solution of the combined decants and colourless crystals. The yellow solution was then concentrated *in vacuo* to give a yellow oil, and the colourless crystals and yellow oil were then analysed separately (see Results and discussion). The crystals were found to have a melting point of 51–53°C, in excellent agreement with that reported by Elming & Clauson-Kaas [13] after recrystallization from methanol (51–52°C).

## 2.5. Electrosynthesis of 2,5-dibutoxy-2,5-dihydrofuran

Sodium butyrate (4.00 g, 36.3 mmol) was suspended in acetonitrile (40 ml), and 60 ml butyric acid was added. The mixture was stirred until all the solids were dissolved, then transferred to the electrochemical cell described in figure 2, cooled to 0°C in an ice bath and the solution degassed for 1 h using a constant stream of nitrogen. After this time, furan (0.50 ml, 0.468 g, 6.88 mmol) was injected into the electrolyte under nitrogen. The constant flow of nitrogen was then immediately switched to a static atmosphere to limit furan evaporation. Bulk electrolysis was then initiated at a potential of 3 V (versus Pt), and the charge passed was monitored. After bulk electrolysis, the pH of the reaction mixture was adjusted to pH 8 with a 1 M solution of sodium carbonate and extracted from aqueous solution with an equal volume of dichloromethane. The organic phase was then washed with water to remove any residual butyrate salts. Concentration of the organic phase *in vacuo* resulted in a viscous brown oil. This oil was added to an equal volume of diethyl ether, precipitating a dark brown solid (which was filtered off and discarded) and a yellowish solution. Concentration of this solution *in vacuo* yielded a yellow oil. This was passed through a silica plug (eluting in dichloromethane) and the solvent removed *in vacuo* to give a pale yellow oil; the ${}^1$H NMR spectrum of this oil showed it to consist of a mixture of the *cis* and *trans* isomers of 2,5-dibutoxy-2,5-dihydrofuran. This compound has been reported and characterized before [13] but no NMR data were reported. Hence we can now report these data as follows: ${}^1$H NMR (CDCl$_3$, 400 MHz), $\delta = 7.01$ (s, 2H, *trans*-H$_a$), 6.78 (s, 1.8H, *cis*-H$_a$), 6.24 (s, 1.8H, *cis*-H$_b$), 6.21 (s, 2H, *trans*-H$_b$), 2.34–2.29 (m, ∼8H, ROOC−**CH$_2$**−CH$_2$−CH$_3$ (*cis* and *trans* signals overlap)), 1.71–1.61 (m, ∼8H, ROOC−CH$_2$−**CH$_2$**−CH$_3$ (*cis* and *trans* signals overlap)), 0.97−0.93

(m, ~12H, ROOC−CH$_2$−CH$_2$−**CH$_3$** (*cis* and *trans* signals overlap)). Letter codes for the protons on the furan ring correspond to those shown in figure 1. $^{13}$C NMR (CDCl$_3$, 125 MHz), $\delta$ = 172.5 and 172.4 (*cis* and *trans* **C**=O), 131.3 and 131.0 (*cis* and *trans* **C**=C in furan ring), 101.4 and 99.8 (*cis* and *trans* sp$^3$ **C** in furan ring), 36.1 and 36.0 (*cis* and *trans* ROOC−**CH$_2$**−CH$_2$−CH$_3$), 18.1 and 18.0 (*cis* and *trans* ROOC−CH$_2$−**CH$_2$**−CH$_3$), 13.5 (×2) (*cis* and *trans* −**CH$_3$**). These NMR spectra are shown in the electronic supplementary material, figures S8 and S9.

## 2.6. Calculations

The program package ORCA was used for all calculations [21]. The input geometry for all molecules was generated using ArgusLab [22]. The geometries of all molecules were fully optimized by a spin-unrestricted density functional theory (DFT) method employing the BP86 functional with acetonitrile as solvent [23,24]. Triple-$\zeta$-quality basis sets with one set of polarization functions (def2-TZVP) were used for all atoms [25,26]. The RIJCOSX approximation combined with the appropriate auxiliary basis set was used to speed up the calculations [27–29]. The conductor-like screening model was used for all calculations [30–32]. The self-consistent field calculations were tightly converged ($1 \times 10^{-8}$ $E_h$ in energy, $1 \times 10^{-7}$ $E_h$ in the charge density and $1 \times 10^{-7}$ in the maximum element of the direct inversion in the iterative subspace [33,34] error vector). The geometry was converged with the following convergence criteria: change in energy less than $10^{-5}$ $E_h$, average force less than $5 \times 10^{-4}$ $E_h$ Bohr$^{-1}$ and the maximum force $10^{-4}$ $E_h$ Bohr$^{-1}$. The geometry search for all complexes was carried out in redundant internal coordinates without imposing geometry constraints. Single-point calculations were performed on optimized coordinates using the B3LYP functional using the same regime of basis sets and solvent [35,36]. The stability of all solutions was checked by performing frequency calculations: no negative frequencies were observed. Spin density plots were obtained using Molekel [37].

## 2.7. Crystallography

Crystallographic data were collected at the University of Glasgow, UK, on a Bruker D8 VENTURE diffractometer equipped with a Photon II CPAD detector, with an Oxford Cryosystems N-Helix device mounted on an I$\mu$S 3.0 (dual Cu and Mo) microfocus sealed tube generator. A colourless, block-shaped crystal of dimensions $0.32 \times 0.20 \times 0.11$ mm was used for single crystal X-ray diffraction data collection. C$_8$H$_{10}$O$_5$ crystallized in the monoclinic space group $P2_1/c$, with unit cell dimensions $a = 11.7447$ (17), $b = 9.7503$ (12), $c = 7.8734$ (10), $\beta = 96.676$ (5)$^\circ$ and $V = 895.5$ (2) Å$^3$, $T = 150$ K. A total of 6125 reflections were measured by $\omega$ scans, 1631 independent reflections with $R_{int} = 0.025$, $\theta_{max} = 25.3^\circ$, $\theta_{min} = 2.7^\circ$ using Mo $K\alpha$ radiation, $\lambda = 0.71073$ Å. The structure was solved using SHELXT [38] and refined using SHELXL [39] (both within OLEX2 [40]). OLEX2 was also used for molecular graphics and to prepare material for publication. Cambridge Crystallographic Data Centre (CCDC) 1895453 contains the supplementary crystallographic data for this paper. More details on the crystallographic data and their collection can be found in the electronic supplementary material.

# 3. Results and discussion

## 3.1. Chemical synthesis and characterization of 2,5-diacetoxy-2,5-dihydrofuran

Before exploring the electrochemical synthesis of 2,5-diacetoxy-2,5-dihydrofuran, we first undertook its chemical synthesis using lead acetate according to the method of Holzapfel & Williams [4] (itself an adaptation of the procedure described by Elming & Clauson-Kaas [13]). This provided us with a benchmark against which to compare the isomeric ratios of our electrochemical experiments, as the relative ratios of the two isomers can be readily discerned by $^1$H NMR (see below). These experiments also provided sufficient quantities of material to allow the isolation of crystals of suitable quality for single crystal X-ray diffraction (see Experimental section).

Figure 3 shows unambiguously that the crystalline form of 2,5-diacetoxy-2,5-dihydrofuran is the *cis* isomer, in which the two acetate moieties have added to the same face of the furan ring. In the product, the five-membered ring itself remains reasonably unpuckered (O1—C2—C3—C4 = 8.05(18)$^\circ$, C2—O1—C5—C4 = 13.59(16)$^\circ$, C2—C3—C4—C5 = 0.26(18)$^\circ$, C3—C4—C5—O1 = 8.45(17)$^\circ$ and C5—O1—C2—C3 = 13.45(16)$^\circ$) with a short C3—C4 interaction of 1.312(2) Å consistent with a carbon–carbon double

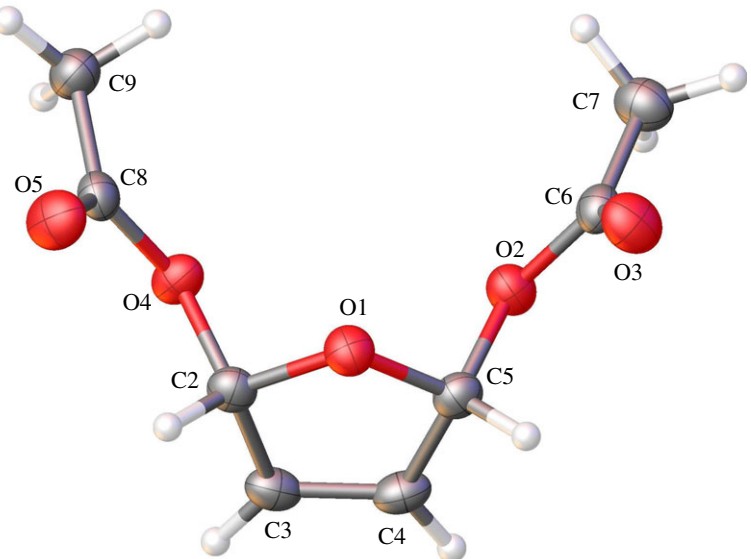

**Figure 3.** The crystal structure of *cis*-2,5-diacetoxy-2,5-dihydrofuran. Crystallographic details can be found in the electronic supplementary material. Colour scheme: C, grey; O, red; H, white. Atomic displacement ellipsoids drawn at 50% probability level.

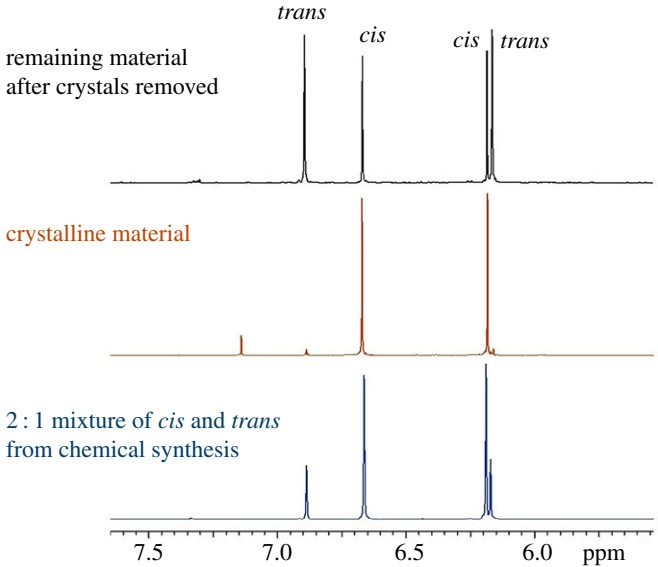

**Figure 4.** Stacked partial $^1$H NMR spectra (400 MHz, 298 K, CDCl$_3$) of the outcome of a typical chemical synthesis of 2,5-diacetoxy-2,5-dihydrofuran (bottom spectrum), the crystalline material (*cis*-2,5-diacetoxy-2,5-dihydrofuran, middle spectrum) and the remaining product material after removal of the crystals (top spectrum).

bond. Meanwhile, the bond lengths between C2 and C3, and between C4 and C5 (1.494(2) Å and 1.490(2) Å, respectively), are both much closer to those typical of carbon–carbon single bonds. The bond angles between the acetate substituents and the ring (O1—C2—O4 = 109.18(12)° and O4—C2—C3 = 107.46(13)° for one acetate and O1—C5—O2 = 110.14(12)° and O2—C5—C4 = 106.87(13)° for the other) suggest a largely tetrahedral geometry around carbons C2 and C5. All this is consistent with the structure of the *cis* isomer drawn in figure 1, where the furan ring has been oxidized and two acetate units have added to the same side of the ring through carbon–oxygen bonds in the 2- and 5-positions.

Armed with this information, we were then able to assign the signals in the $^1$H NMR spectrum of isomeric 2,5-diacetoxy-2,5-dihydrofuran definitively for the first time. Figure 4 shows a partial $^1$H NMR stack plot of the outcome of a typical chemical synthesis of 2,5-diacetoxy-2,5-dihydrofuran after removal of the lead salts by filtration and washing out the excess acetic acid by aqueous extraction (bottom spectrum). The middle spectrum shows the $^1$H NMR spectrum of the crystalline material, the structure of which is shown in figure 3, and which therefore corresponds to the *cis* isomer. On the basis of the

chemical shifts shown by this purified isomer, it seems apparent that the two signals at 6.86 and 6.13 ppm arise from the *trans* isomer and the two signals in between the aforementioned correspond to the *cis* isomer. This is borne out in the upper spectrum, which shows the [1]H NMR spectrum of the remaining product material after removal of the crystals; clearly the ratio of the peaks for the *cis* isomer relative to that for the *trans* isomer has decreased in intensity as some of the *cis* isomer has been removed as crystals. An HMQC analysis then allows the two signals for each isomer to be assigned to the relevant protons on the basis of the cross-peaks seen in the associated [1]H and [13]C NMR spectra (electronic supplementary material, figures S5 and S6). Further corroboration is to be gained from an [1]H NMR COSY, which shows that the inner peaks couple to each other (but not to the outer peaks; electronic supplementary material, figure S7). This then leads to the assignments shown in figure 4. With this information in hand, we performed multiple repeats of this chemical synthesis method and (through analysis of the [1]H NMR spectra after removal of acetic acid and lead salts) we were able to show that the *cis* isomer is the dominant product under these chemical conditions, forming in a ratio of *cis* : *trans* of $1.88(\pm 0.03):1$. This ratio is in very good agreement with that reported by Holzapfel & Williams [4] (2 : 1 major : minor), but we note that these authors (and others [17,18]) previously (incorrectly) assumed the major isomer to be the *trans* isomer on the basis of NMR coupling constants and mechanistic considerations. However, by isolating the *cis* form and solving its crystal structure, we show that in fact the *cis* form is the major isomer.

## 3.2. Electrochemical synthesis of 2,5-diacetoxy-2,5-dihydrofuran

With the ability to now obtain the ratios of the two isomers of 2,5-diacetoxy-2,5-dihydrofuran from integration of the appropriate peaks in the [1]H NMR spectrum, we were able to study the *cis*–*trans* ratio produced by electrochemical oxidation of furan in the presence of acetic acid for the first time. Accordingly, a 0.5 M solution of sodium acetate in 4 : 1 acetic acid : acetonitrile was cooled to $0^{\circ}$C in an ice bath and then charged with furan to give a 0.07 M solution of furan in the electrolyte (see Experimental section). Bulk electrolysis was then conducted on this solution in a single chamber cell using a large surface area reticulated vitreous carbon working electrode, a graphite rod counter electrode and a Pt wire pseudo-reference electrode at an applied potential of $+3$ V (versus Pt). The ice bath was necessary to slow down the rate of furan evaporation (b.p. $= 31^{\circ}$C), so that meaningful Faradaic yields could be estimated. Even so, we found it convenient to limit the total charge passing during these experiments to no more than 30% of that required to produce full conversion of the furan. This not only enabled experiments of short duration to be performed (reducing anomalies due to furan evaporation) but also provided some useful insights into the mechanism of furan oxidation (see below).

In a typical bulk electrolysis, 0.5 ml (0.468 g, 6.88 mmol) of furan was dissolved in 100 ml of electrolyte and subjected to an anodic potential of $+3$ V versus Pt (see electronic supplementary material, figure S10 for a cyclic voltammogram) until 109 C had been passed. This corresponds to about 8% of the total charge required to oxidize this amount of furan by two electrons. Using Faraday's law, this in turn equates to a maximum theoretical yield of 2,5-diacetoxy-2,5-dihydrofuran of 0.56 mmol (assuming a two-electron oxidation process). In the event, the combined yields of the crystals (pure *cis* isomer) and oil (a mixture of *cis* and *trans* isomers) from this experiment was 83 mg (0.45 mmol), corresponding to a Faradaic yield of 80%.

Multiple repeats of the above reaction were performed passing various amounts of charge. In all cases, after bulk electrolysis had been terminated, the reaction mixture was poured into a large volume of water and extracted with dichloromethane. Repeated washing of the dichloromethane phase removed the vast majority of the excess acetate salts. After concentration under reduced pressure, the resulting brown oil was added to an equal volume of diethyl ether, giving a dark brown solid and yellow solution. Concentration of this solution *in vacuo* yielded a yellow oil, the [1]H NMR spectrum of which showed it to consist almost exclusively of a mixture of the *cis* and *trans* isomers of 2,5-diacetoxy-2,5-dihydrofuran. The ratio of *trans* and *cis* isomers was then judged on the basis of the integration of these NMR spectra, giving a *cis* : *trans* ratio of $1.38(\pm 0.18):1$, or 7 : 5 *cis* : *trans*. Hence the electrochemical route appears to be a lot less selective towards the *cis* product than the chemical route (the latter gives approx. 2 : 1 *cis* : *trans*). Table 1 shows a comparison of our electrochemical conditions and results with those from previous electrosynthesis studies.

## 3.3. Insights into the mechanism of 2,5-diacetoxy-2,5-dihydrofuran electrosynthesis

During the course of these studies, an intriguing observation was made: despite the fact that the charges that were passed were typically only sufficient to bring about a partial oxidation of the furan, no evidence

**Scheme 1.** The ECEC mechanism proposed by Yoshida & Fueno [41] and Atobe and co-workers [42] for the electrosynthesis of 2,5-dimethoxy-2,5-dihydrofuran.

**Table 1.** A comparison of the previous electrochemical routes for the synthesis of 2,5-diacetoxy-2,5-dihydrofuran and the results from this paper.

| no. | anode | cathode | cell voltage | Faradaic efficiency (%) | cis : trans ratio | conditions | ref. |
|---|---|---|---|---|---|---|---|
| 1 | Pt | Pt | AC electrolysis | — | — | batch[a] | [14] |
| 2 | Pt | Hg | 100 V DC | — | — | batch[b] | [15] |
| 3 | Pt | C | — | 46 | — | batch[c] | [7] |
| 4 | graphite | Pt | — | 40[d] | — | flowcell[e] | [16] |
| 5 | reticulated carbon | graphite | 3 V versus Pt | 80 | 7 : 5 | batch[c] | this work |
| 6[f] | reticulated carbon | graphite | 3 V versus Pt | 45 | 9 : 10 | batch | this work |

[a]In 1 M potassium acetate in acetic acid.
[b]In 0.6 M sodium acetate in acetic acid.
[c]In 0.5 M sodium acetate in 4 : 1 acetic acid : acetonitrile.
[d]At a flow rate of 1 ml min$^{-1}$.
[e]In acetic acid without additional electrolyte.
[f]For the electrosynthesis of 2,5-dibutoxy-2,5-dihydrofuran from 0.36 M sodium butyrate in a 2 : 3 (by volume) solution of acetonitrile and butyric acid.

for the generation of any mono-substituted furans (such as a putative 2-acetoxy-2,5-dihydrofuran) was ever observed. This was even the case if samples of electrolyte from ongoing bulk electrolyses were extracted and examined by $^1$H NMR spectroscopy without any work-up or concentration under reduced pressure. This in turn suggested that the mechanism of oxidation strongly favoured a second oxidation reaction after the first acetate had added, or indeed that the addition of both acetate groups might be concerted.

In this context, both Yoshida [41] and Atobe [42] and their co-workers have proposed that the somewhat related electrosynthesis of dimethoxylated furans by electro-oxidation of furan occurs by an electrochemical–chemical–electrochemical–chemical (ECEC) mechanism, as shown in scheme 1. Hence furan is first oxidized at the electrode to give a radical cation, which then reacts with the first methoxy anion to give the mono-substituted radical species. A second electrochemical oxidation of this radical at the anode produces a closed-shell cation which undergoes attack by the second methoxy anion to generate the dimethoxylated product.

To place this proposed mechanism in a more formal context, we performed DFT calculations on the various likely intermediate species that would form in the analogous ECEC pathway for the electro-synthesis of 2,5-diacetoxy-2,5-dihydrofuran, and these results are summarized in scheme 2. Hence the initial one-electron oxidation of furan is significantly endergonic (by 126.5 kcal mol$^{-1}$) and produces a radical cation where the radical and cation can be considered to reside predominantly at the 2- and 5-positions, in agreement with the suggested mechanism in scheme 1. A direct two-electron oxidation of furan can be ruled out on the basis of the very high energy barrier (314.9 kcal mol$^{-1}$) to the formation of the dication (grey lines in scheme 2). Hence a stepwise pathway is likely to prevail.

After the formation of the radical cation, acetate then adds to the cation, leading to a stabilization of the system of around 10 kcal mol$^{-1}$. A second electrochemical oxidation reaction then occurs to generate

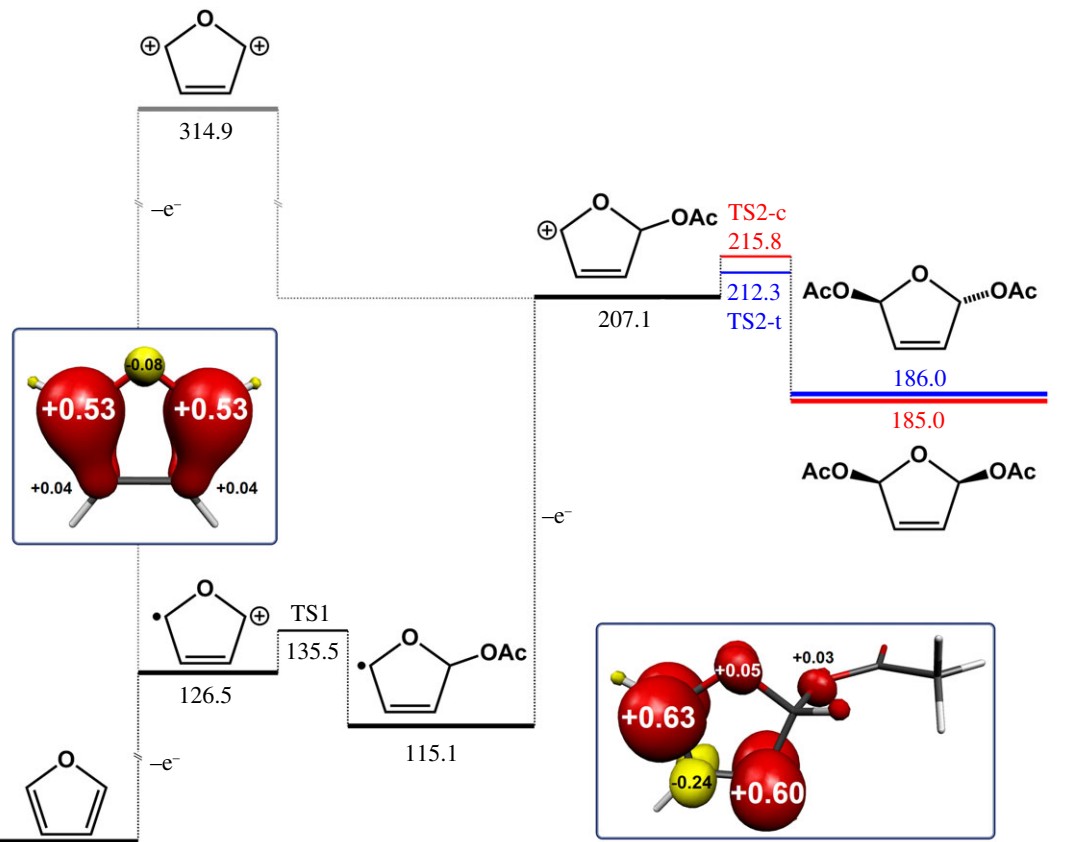

**Scheme 2.** The proposed reaction pathway for the electrosynthesis of 2,5-diacetoxy-2,5-dihydrofuran computed at the B3LYP level of theory. The red horizontal lines refer to *cis*-2,5-diacetoxy-2,5-dihydrofuran, and the blue horizontal lines to *trans*-2,5-diacetoxy-2,5-dihydrofuran. All values are given in kcal mol$^{-1}$. Insets show the Mulliken spin population analysis for the radical cation and mono-substituted neutral radical (red: $\alpha$-spin; yellow: $\beta$-spin).

a cation at the 5-position of the furan ring, where stabilization by the formation of an oxonium ion is possible. This second oxidation is considerably less endergonic than the first oxidation reaction (by over 34 kcal mol$^{-1}$), suggesting that this second oxidation reaction should proceed very rapidly after the first acetate has added to the furan, provided that the anodic bias continues to be supplied. This in turn offers an explanation for the absence of any mono-substituted products, as the neutral mono-substituted radical is easier to oxidize than the starting material, and hence rapidly undergoes a second oxidation once formed. This cation then reacts with a second acetate anion to generate the di-substituted products. The energies predicted for the transition states leading to the *cis* and *trans* products (and indeed the energies predicted for the *cis* and *trans* products themselves) are within the 3–5 kcal mol$^{-1}$ error usual for DFT and are therefore not considered to be significantly different from each other in this analysis.

## 3.4. Electrochemical synthesis of 2,5-dibutoxy-2,5-dihydrofuran

If correct, then the above mechanism implies that the second transition state (TS2) is critical for determining the stereochemistry of the resulting products. Molecular representations of the closed-shell cationic intermediate immediately preceding this transition state (manipulated and visualized using the program Avogadro [43,44]; figure 5) suggest that addition of a particularly bulky carboxylate group in the first addition step could have an influence on the *cis* : *trans* ratio of the products by virtue of that bulky group partially blocking the face to which it adds to further nucleophilic attack. The expected result of this would be an increase in the amount of *trans* product produced relative to the *cis*, as nucleophilic addition to the opposite face of the furan ring would be less sterically challenged.

To test this hypothesis, we conducted the electrochemical oxidation of furan (0.07 M) in an electrolyte consisting of 0.36 M sodium butyrate in a 2 : 3 (by volume) solution of acetonitrile and butyric acid. All

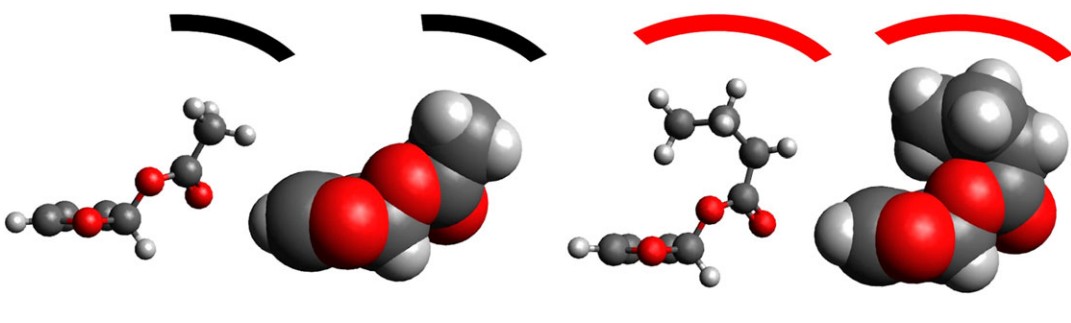

top face only partly occluded by first substituent

top face significantly occluded by first substituent

acetate substituent

butyrate substituent

**Figure 5.** Wire and stick and space-filling models showing the effect that a bulky substituent (butyrate, right) has on the shielding of one face of the furan ring to the addition of a second carboxylate moiety, compared with the much smaller acetate substituent (left). The butyrate substituent is shown in a particularly shielding conformation; other conformations are similar in energy but less shielding. However, it is to be expected that the conformation shown (and other very shielding conformations similar to it) will be adopted in solution for a not insignificant proportion of the time.

other parameters were kept as close as possible to those employed using the acetate/acetic acid electrolyte (see Experimental section). In a typical bulk electrolysis, 0.5 ml (0.468 g, 6.88 mmol) of furan was dissolved in 100 ml of this electrolyte and subjected to an anodic potential of $+3$ V (versus Pt) until 396 C had been passed. This corresponds to 30% of the total charge required to oxidize this amount of furan by two electrons. Using Faraday's law, this in turn equates to a maximum theoretical yield of 2,5-dibutoxy-2,5-dihydrofuran of 2.05 mmol (assuming a two-electron oxidation process). In the event, the combined yield of *cis* and *trans* isomers from this experiment after work-up and purification was 222 mg (0.92 mmol), corresponding to a Faradaic yield of 45%. This lower Faradaic yield for isolated 2,5-dibutoxy-2,5-dihydrofuran relative to 2,5-diacetoxy-2,5-dihydrofuran is likely to be due to greater losses for 2,5-dibutoxy-2,5-dihydrofuran during the work-up and purification steps (for example, the need for column chromatography). The isomers of 2,5-dibutoxy-2,5-dihydrofuran were obtained in a ratio of *cis*:*trans* of $0.88(\pm0.1):1$, indicating a distinct reversal in the stereoselectivity of the electrochemical synthesis from 7:5 in favour of *cis* when using acetate to 10:9 in favour of *trans* when using the bulkier butyrate substituent. This suggests that the stepwise ECEC mechanism proposed for the electrosynthesis of 2,5-diacetoxy-2,5-dihydrofurans in scheme 2 is indeed operating.

## 4. Conclusion

In conclusion, we have solved the crystal structure of *cis*-2,5-diacetoxy-2,5-dihydrofuran for the first time, which has allowed us to assign the two isomers of 2,5-diacetoxy-2,5-dihydrofuran to their respective $^1$H NMR spectra. This has led to the realization that the previous assignments of the $^1$H NMR spectra of *cis* and *trans*-2,5-diacetoxy-2,5-dihydrofuran were incorrect. As a result of this discovery, we have been able to give reliable information on the ratio of these isomers that form during the electrochemical oxidation of furan in acetate electrolytes, also for the first time (a ratio of 7:5 *cis*:*trans* under our conditions). A mechanism for this reaction consistent with our experimental observations was proposed and then partially validated by conducting an analogous electrosynthesis of 2,5-dibutoxy-2,5-dihydrofuran using the much bulkier butyrate substituent. As predicted by the postulated mechanism, the greater steric hindrance led to a shift in stereoselectivity in favour of the *trans* isomer, resulting in a *cis*:*trans* ratio of 9:10. The implications of these findings for controlling product distributions in other electrosynthesis reactions on furan substrates are currently under investigation in our laboratories.

Data accessibility. The data which underpin this work are available via Enlighten, the University of Glasgow's open access online data repository, at http://dx.doi.org/10.5525/gla.researchdata.745. CCDC 1895453 contains the supplementary crystallographic data for this paper. These data can be obtained free of charge from the Cambridge Crystallographic Data Centre via www.ccdc.cam.ac.uk/data_request/cif.

Authors' contributions. M.A.S. performed electrochemical measurements and product purification and analysis; S.S. performed the DFT calculations and C.W. collected and analysed the crystallographic data. M.D.S. supervised the work, and M.A.S. and M.D.S. co-wrote the manuscript. All authors gave final approval for publication.

Competing interests. We declare we have no competing interests.

Funding. M.D.S. thanks the Royal Society for a University Research Fellowship (UF150104) and the University of Glasgow for funding.

Acknowledgements. The authors thank Dr David France (University of Glasgow) for useful discussions.

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
