## [Reviewer comments · Royal Society Open Science]

Review History

RSOS-190336.R0 (Original submission)

Review form: Reviewer 1

Is the manuscript scientifically sound in its present form?

Yes

Are the interpretations and conclusions justified by the results?

Yes

Is the language acceptable?

Yes

Is it clear how to access all supporting data?

Yes

Do you have any ethical concerns with this paper?

No

Have you any concerns about statistical analyses in this paper?

No

Recommendation?

Accept with minor revision (please list in comments)

Comments to the Author(s)

The author described an electrochemical synthesis of 2,5-diacetoxy-2,5-dihydrofuran and provided a method for the separation of stereoisomers. This report is useful, because the pure isomer has been obtained and thus the stereoselectivity of the electrochemical reaction was obtained. Furthermore, DFT calculations also were carried out for the understanding of the reaction mechanism. This reviewer recommends acceptance of this manuscript for publication after minor revisions.

1. I suggest to remove "structure" in the title, as this work mainly deals with stereoselectivity and mechanism, while chemical structure is known.
2. Only 2,5-diacetoxy-2,5-dihydrofuran was synthesized in this work, why the R = small or bulky groups were shown in the graphic abstract?

Review form: Reviewer 2

Is the manuscript scientifically sound in its present form?

Yes

Are the interpretations and conclusions justified by the results?

Yes

Is the language acceptable?

Yes

Is it clear how to access all supporting data?

Yes

Do you have any ethical concerns with this paper?

No

Have you any concerns about statistical analyses in this paper?

No

Recommendation?

Accept with minor revision (please list in comments)

Comments to the Author(s)

This study provides new mechanistic insight into a 2-electron oxidation of furan leading to valuable 1,4-substituted products. The chemical process with Pb(IV) is compared to an elegant electrosynthesis process. The mechanism and steric effects on cis-/trans product ratio are considered. The work offers new insights based on several analytical approaches (including theory) and the conclusions are clear and supported by high quality data. Publication is recommended.

The manuscript is well written and structured. Good quality figures support the text. Title and abstract are consistent with the main text. Supporting information are useful. Citation of references is appropriate and sufficient. Some minor revision ideas:

(A) could the counter electrode process in the electrosynthesis be identified?

(B) in table 1, could both acetate and butyrate results be listed?

(C) 80% faradaic efficiency is suggested but for the butyrate only less than 50%. Any suggestions? Could unidentified products be present (e.g. furan ring-opening)?

(D) +3V vs Pt sounds like a very high and somewhat arbitrary choice of potential. Was there any attempt to record CVs or to vary the potential?

(E) was the synthesis always stopped at 10% conversion? Could a better conversion be achieved? Could a plot of yield versus conversion be given?

(F) was succinate considered as a reagent?

Decision letter (RSOS-190336.R0)

28-May-2019

Dear Dr Symes:

Title: Towards a Better Understanding of the Electrosynthesis of 2,5-dicarboxy-2,5-dihydrofurans: Structure, Mechanism and Influence over Stereochemistry

Manuscript ID: RSOS-190336

Thank you for submitting the above manuscript to Royal Society Open Science. On behalf of the Editors and the Royal Society of Chemistry, I am pleased to inform you that your manuscript will be accepted for publication in Royal Society Open Science subject to minor revision in accordance with the referee suggestions. Please find the reviewers' comments at the end of this email. I apologise that this has taken longer than usual.

The reviewers and handling editors have recommended publication, but also suggest some minor revisions to your manuscript. Therefore, I invite you to respond to the comments and revise your manuscript.

Because the schedule for publication is very tight, it is a condition of publication that you submit the revised version of your manuscript before 06-Jun-2019. Please note that the revision deadline will expire at 00.00am on this date. If you do not think you will be able to meet this date please let me know immediately.

Best wishes,
Dr Laura Smith
Publishing Editor, Journals

On behalf of the Subject Editor Professor Anthony Stace and the Associate Editor Professor John Moses.

RSC Associate Editor:
Comments to the Author:

(There are no comments.)

RSC Subject Editor:

Comments to the Author:

(There are no comments.)

Reviewer comments to Author:

Reviewer: 1

Comments to the Author(s)

The author described an electrochemical synthesis of 2,5-diacetoxy-2,5-dihydrofuran and provided a method for the separation of stereoisomers. This report is useful, because the pure isomer has been obtained and thus the stereoselectivity of the electrochemical reaction was obtained. Furthermore, DFT calculations also were carried out for the understanding of the reaction mechanism. This reviewer recommends acceptance of this manuscript for publication after minor revisions.

1. I suggest to remove "structure" in the title, as this work mainly deals with stereoselectivity and mechanism, while chemical structure is known.
2. Only 2,5-diacetoxy-2,5-dihydrofuran was synthesized in this work, why the R = small or bulky groups were shown in the graphic abstract?

Reviewer: 2

Comments to the Author(s)

This study provides new mechanistic insight into a 2-electron oxidation of furan leading to valuable 1,4-substituted products. The chemical process with Pb(IV) is compared to an elegant electrosynthesis process. The mechanism and steric effects on cis-/trans product ratio are considered. The work offers new insights based on several analytical approaches (including theory) and the conclusions are clear and supported by high quality data. Publication is recommended.

The manuscript is well written and structured. Good quality figures support the text. Title and abstract are consistent with the main text. Supporting information are useful. Citation of references is appropriate and sufficient. Some minor revision ideas:

(A) could the counter electrode process in the electrosynthesis be identified?

(B) in table 1, could both acetate and butyrate results be listed?

(C) 80% faradaic efficiency is suggested but for the butyrate only less than 50%. Any suggestions? Could unidentified products be present (e.g. furan ring-opening)?

(D) +3V vs Pt sounds like a very high and somewhat arbitrary choice of potential. Was there any attempt to record CVs or to vary the potential?

(E) was the synthesis always stopped at 10% conversion? Could a better conversion be achieved? Could a plot of yield versus conversion be given?

(F) was succinate considered as a reagent?

Author's Response to Decision Letter for (RSOS-190336.R0)

See Appendix A.

Decision letter (RSOS-190336.R1)

10-Jun-2019

Dear Dr Symes:

Title: Towards a Better Understanding of the Electrosynthesis of 2,5-dicarboxy-2,5-dihydrofurans: Structure, Mechanism and Influence over Stereochemistry
Manuscript ID: RSOS-190336.R1

It is a pleasure to accept your manuscript in its current form for publication in Royal Society Open Science. The chemistry content of Royal Society Open Science is published in collaboration with the Royal Society of Chemistry.

On behalf of the Subject Editor Professor Anthony Stace.

RSC Associate Editor
Comments to the Author:
The referees' comments have been addressed sufficiently and the manuscript can now be accepted as is.

Reviewer(s)' Comments to Author:

Appendix A

We thank the referees for their careful reading of our manuscript and for the useful suggestions that they have made. In the following, we respond to all the referees' comments (shown in italics) in turn (our answers are in bold type). Where we have made changes to the manuscript and SI in response to these changes, we have highlighted these in yellow. We believe that the changes made to the manuscript have improved it significantly, and we thank the referees for their help in this.

Reviewer: 1

The author described an electrochemical synthesis of 2,5-diacetoxy-2,5-dihydrofuran and provided a method for the separation of stereomers. This report is useful, because the pure isomer has been obtained and thus the stereoselectivity of the electrochemical reaction was obtained. Furthermore, DFT calculations also were carried out for the understanding of the reaction mechanism. This reviewer recommends acceptance of this manuscript for publication after minor revisions.

1. I suggest to remove "structure" in the title, as this work mainly deals with stereoselectivity and mechanism, while chemical structure is known.

The work also reports the crystal structure of *cis*-2,5-diacetoxy-2,5-dihydrofuran for the first time, and the information provided by this structure is central to correctly interpreting the NMR spectra of the species produced by the electrosynthesis reactions. Therefore, we prefer to leave "structure" in the title to emphasise this.

2. Only 2,5-diacetoxy-2,5-dihydrofuran was synthesized in this work, why the R = small or bulky groups were shown in the graphic abstract?

Whilst only 2,5-diacetoxy-2,5-dihydrofuran was isolated and separated into its stereoisomers, 2,5-dibutyl-2,5-dihydrofuran was also synthesised as a mixture of *cis* and *trans* isomers (whose ratios we determine). Therefore, we consider the graphical abstract to be appropriate.

Reviewer 2:

*This study provides new mechanistic insight into a 2-electron oxidation of furan leading to valuable 1,4-substituted products. The chemical process with Pb(IV) is compared to an elegant electrosynthesis process. The mechanism and steric effects on *cis*-/*trans* product ratio are considered. The work offers new insights based on several analytical approaches (including theory) and the conclusions are clear and supported by high quality data. Publication is recommended.*

The manuscript is well written and structured. Good quality figures support the text. Title and abstract are consistent with the main text. Supporting information are useful. Citation of references is appropriate and sufficient. Some minor revision ideas:

(A) could the counter electrode process in the electrosynthesis be identified?

At this stage, we have not yet studied the reactions at the counter electrode in any detail. We used a three-electrode set-up for our experiments, so the potential at the counter electrode is likely to be large and negative. Under these conditions, the prevailing reactions are likely to be degradation of the species most abundant in solution (solvent and supporting electrolyte). However, we cannot speculate further at this stage.

(B) in table 1, could both acetate and butyrate results be listed?

The butyrate results are now listed in this table as new entry 6, along with a brief note on conditions in the table footnotes.

(C) 80% faradaic efficiency is suggested but for the butyrate only less than 50%. Any suggestions? Could unidentified products be present (e.g. furan ring-opening)?

Our Faradaic yields are based on the isolated masses of purified products free of any other species (such as solvents and electrolyte salts). Most problematic in this case is the removal of the acetate or butyrate salts. For the acetic acid/acetate mixes, washing with water and concentration under reduced pressure removed the acetate/acetic acid rather effectively. However, in the case of butyrate/butyric acid, washing with water was only partially effective and (as we say in the Experimental section) we had to column our product material to fully remove these species. This also caused considerable loss of product material, which is the main reason for the apparently diminished Faradaic yield in this case. A statement to this effect has been added to the main text where we discuss the Faradaic yield for 2,5-dibutoxy-2,5-dihydrofuran.

(D) +3V vs Pt sounds like a very high and somewhat arbitrary choice of potential. Was there any attempt to record CVs or to vary the potential?

We did indeed record a cyclic voltammogram and this is now shown in Figure S10 in the supporting information. We also call out this new figure where we first discuss the potential we chose for the oxidation reaction. There is a broad (irreversible) oxidation peak at around +2.5 V vs. Pt in the CV. Oxidations were run at potentials somewhat higher than this to overcome solution resistance and to speed up the rate of oxidation.

(E) was the synthesis always stopped at 10% conversion? Could a better conversion be achieved? Could a plot of yield versus conversion be given?

As we mention in the main text on page 14, in the results that we report we limited the total charge passing during these experiments to no more than 30% of that required to produce full conversion of the furan. We did this because (as we also mention) even when cooled in an ice bath, furan is still rather volatile in our experimental set-up and if we attempted higher conversions (which would require longer experiments) conversions suffered due to this loss of starting material. Faradaic yields also suffered as a result. Hence, we did experiment with other conversions, but found that 30% represented a good compromise between a fair level of overall conversion without the issues of starting material evaporation becoming significant. Better conversions might be possible with an improved experimental set-up (more effective cooling, or the use of a flow-cell arrangement) and we will explore these options more thoroughly in future work.

(F) was succinate considered as a reagent?

We did not (yet) investigate succinate as a reagent. However, this is a most interesting suggestion for future work because succinate has two carboxylic acid groups and so there would be the possibility of forming polymeric materials or (given the short chain length in succinate) also the possibility of intramolecular reactions. The ratio of *cis* to *trans* isomers in the latter case would be very interesting to probe. We thank the referee for this suggestion.